# Light- and Melanin Nanoparticle-Induced Cytotoxicity in Metastatic Cancer Cells

**DOI:** 10.3390/pharmaceutics13070965

**Published:** 2021-06-26

**Authors:** Victoria R. Gabriele, Robabeh M. Mazhabi, Natalie Alexander, Purna Mukherjee, Thomas N. Seyfried, Njemuwa Nwaji, Eser M. Akinoglu, Andrzej Mackiewicz, Guofu Zhou, Michael Giersig, Michael J. Naughton, Krzysztof Kempa

**Affiliations:** 1Department of Physics, Boston College, Chestnut Hill, MA 02467, USA; gabrievi@bc.edu (V.R.G.); naughton@bc.edu (M.J.N.); 2International Academy of Optoelectronics at Zhaoqing, South China Normal University, Zhaoqing 526238, China; r.motaghed@zq-scnu.org (R.M.M.); njemuwa.nwaji@zq-scnu.org (N.N.); e.a@fu-berlin.de (E.M.A.); guofu.zhou@m.scnu.edu.cn (G.Z.); giersig@physik.fu-berlin.de (M.G.); 3Department of Biology, Boston College, Chestnut Hill, MA 02467, USA; njalexander343@gmail.com (N.A.); purna.mukherjee@bc.edu (P.M.); thomas.seyfried@bc.edu (T.N.S.); 4Greater Poland Cancer Centre, Poznan University of Medical Sciences, 61-867 Poznan, Poland; mackiewicz.aa@gmail.com; 5Guangdong Provincial Key Laboratory of Optical Information Materials and Technology & Institute of Electronic Paper Displays, South China Academy of Advanced Optoelectronics, South China Normal University, Guangzhou 510006, China; 6Institute of Fundamental Technological Research of Polish Academy of Sciences, 02-106 Warsaw, Poland

**Keywords:** melanoma, melanin nanoparticles, cytotoxicity, laser medical applications, hyperthermia

## Abstract

Melanin nanoparticles are known to be biologically benign to human cells for a wide range of concentrations in a high glucose culture nutrition. Here, we show cytotoxic behavior at high nanoparticle and low glucose concentrations, as well as at low nanoparticle concentration under exposure to (nonionizing) visible radiation. To study these effects in detail, we developed highly monodispersed melanin nanoparticles (both uncoated and glucose-coated). In order to study the effect of significant cellular uptake of these nanoparticles, we employed three cancer cell lines: VM-M3, A375 (derived from melanoma), and HeLa, all known to exhibit strong macrophagic character, i.e., strong nanoparticle uptake through phagocytic ingestion. Our main observations are: (i) metastatic VM-M3 cancer cells massively ingest melanin nanoparticles (mNPs); (ii) the observed ingestion is enhanced by coating mNPs with glucose; (iii) after a certain level of mNP ingestion, the metastatic cancer cells studied here are observed to die—glucose coating appears to slow that process; (iv) cells that accumulate mNPs are much more susceptible to killing by laser illumination than cells that do not accumulate mNPs; and (v) non-metastatic VM-NM1 cancer cells also studied in this work do not ingest the mNPs, and remain unaffected after receiving identical optical energy levels and doses. Results of this study could lead to the development of a therapy for control of metastatic stages of cancer.

## 1. Introduction

The emergence of nanoparticle (NP) technology in biomedicine has led to many applications [1,2]. These include tumor imaging and targeting [3], tissue engineering [4], drug delivery [5], tumor destruction [6], pathogen detection [7], and protein detection [8], among others. Sufficiently small nonpolar NPs can cross biological barriers and translocate across cells, tissues, and organs [9]. In contrast, polar NPs can enter cells only by utilizing endocytotic pathways [10,11]. The internalization process of NPs by cells is a key factor in determining their biomedical function, toxicity, and biodistribution [11]. Adjusting chemophysical properties of NPs, such as size, shape, and surface properties, is a major factor for optimization of targeting and cellular uptake, as well as intracellular trafficking [12]. Zeta potential (ξ) could be an important biophysical parameter for quantification of the cellular interactions [13,14,15].

Meanwhile, it has been also known for over a century that biomolecules can be irreversibly damaged by *ionizing radiation*, via photons with energy sufficient to break covalent bonds. For example, ultraviolet (UV) radiation is known to cause catastrophic damage to cells [16], and X-rays and even harder radiation have been long applied to treat cancer [17]. Such radiation damages of cells and tissue is largely indiscriminate, with minimal or no spectral control or biospecificity. Therefore, geometric targeting must be used to achieve some degree of macro-scale selectivity. *Non-ionizing radiation*, with photons of much lower energy can, at sufficient intensity, also cause irreversible damage to biomolecules via nonlinear processes (under high local electric or thermal field). Geometric targeting can be improved with such radiation due to the availability of lensing, in particular in the visible frequency range. Irreversible damage of geometrically microtargeted yeast cells was recently demonstrated, using laser tweezers employing a low power (80 mW), near infrared (NIR) laser focused to a spot of about 1 µm diameter (~10^10^ W/m^2^) [18]. Most importantly, however, nonlinear effects produced by non-ionizing radiation allow for spectral resolution of the excitation. Spectra in the NIR and far IR (FIR) ranges consist typically of characteristic groups of absorbance maxima, which form so called “fingerprint” spectra, and which can be used to identify a given molecule [19]. A recent theoretical paper [20] suggested that such fingerprint spectra can be used to selectively damage target molecules within a cell. Such purely spectral selectivity of molecular dissociation would be highly desirable in future therapies, but it is currently very hard (or impossible) to achieve/implement, mainly because the spectra of different biomolecules (ranging from viral to cellular, healthy or cancerous) are very similar, typically with only some amplitude variations, but at similar or the same peak spectral locations (wavelengths) [21]. An additional complication is the generally small radiation penetration depth, apart from a few high transparency spectral windows [21,22].

These technical difficulties can be overcome with the incorporation of strongly light-absorbing targets, such as NPs. For example, light absorption by melanin NPs is very strong (typically an order of magnitude more than typical cells) over a wide spectral range, a fact that has been exploited in the detection of metastatic melanoma circulating tumor cells (CTC) [23,24]. Several papers [23,25] have shown that, for wavelengths around 500 nm and between 700 nm and 900 nm, melanoma cells dominate absorption over that of blood, suggesting they may be able to be overheated with radiation at those wavelengths. In fact, melanin-filled NPs have been used recently to trigger cell death by overheating (over 42 °C) in tumors [26]. In such tumor therapy, radiation in the NIR high transparency window (~800 nm wavelength) is typically used.

We have developed highly monodispersed glucose-coated melanin nanoparticles (mNP@G), and have used these to reveal massive NP uptake by the three cancer cell lines, VM-M3, A375, and HeLa, which confirm these cell types’ macrophagic character. Zeta potential measurements suggest that this character is related to binding and cellular internalization effects. Importantly, we find that the viability of all studied cells dramatically decreases at a sufficiently high concentration of mNP@G and reduction of the glucose level in the culture nutrition. We also performed a series of radiation experiments on cancer cells moderately filled with mNP@G. We employed light in the visible transmission window of blood at 532 nm wavelength [22,23], and demonstrated that there exist power levels and doses of this radiation that violently destroy cancer cells sensitized with mNPs, but that are evidently safe for cells unsensitized with mNPs.

While melanin, the pigment present in abundance in melanoma cells, plays an important role in skin protection against ultraviolet radiation, it also affects melanoma behavior by adjusting epidermal homeostasis [27,28]. Melanoma is, of course, a serious skin cancer, originating from mutated melanocytes, melanin-producing cells [29]. Highly metastatic, it causes about 60,000 deaths per year globally [30]. Very limited progress treating melanoma has been achieved with chemotherapy [31], immunotherapy [32], radiotherapy [33], surgery [34,35], or other therapies [36,37]. Melanin synthesis, a multistep and highly regulated route, determines the difference between the function of normal and cancerous cells [38]. Different from healthy melanocytes, in which melanin synthesis is controlled by various factors and plays an important biological role, melanin pigmentation in melanoma cells is dysregulated, which leads to heavy pigmentation of these cells [39,40]. Sarna et al. [41] have suggested that the elastic properties of melanoma cells are affected by the melanin presence, and play a key role in melanoma metastasis [38]. Other studies confirm that melanin pigmentation is an important factor in determining the fate of cancer cells [39,42]. Metabolic functions of normal cells are dramatically changed in the cancerous state, and this transformation makes cancer cells strongly dependent on high rates of glucose uptake [43,44].

To achieve rapid cancer cell proliferation in vitro, cell culturing methods commonly use high glucose of Dulbecco’s modified Eagle’s medium (DMEM, 25 mM or 4500 µg·mL^−1^). Normal serum glucose levels in the body are usually constant between 4 and 6 mM (720–1080 µg·mL^−1^). However, the body may experience a drop in glucose level to 2.5 mM (450 µg·mL^−1^), and even further in tissue, in the case of nutrient deficiencies. Accordingly, glucose level reduction has been applied for cancer treatment through different methods such as fasting or modifying (e.g., ketogenic) diet [45,46].

## 2. Materials and Methods

### 2.1. Reagents

Chemicals were purchased from commercial sources with high purity and used as received. Malignant melanoma A375 and HeLa cell lines were obtained from the Shanghai Institute of Cell Biology (Shanghai, China). The cells were cultured in DMEM (Solarbio, Beijing, China) with 10% fetal bovine serum (Solarbio) and 5% antibiotics (100 Unit mL^−1^ penicillin and 100 µg·mL^−1^ streptomycin) from Sigma-Aldrich (St. Louis, MO, USA). The cells were incubated in a cell incubator under 95% humidity and 5% CO_2_ at 37 °C. To seed and harvest the cells, Trypsin-EDTA (0.25%) from Sigma was utilized, and Trypan blue (0.4%) from Gibco was applied for cell counting purposes.

### 2.2. Synthesis of Melanin Nanoparticles, mNP

The synthesis of highly spherical monodispersed mNPs was accomplished using the oxidative polymerization of dopamine hydrochloride in the presence of ethanol and ammonia solution at room temperature. A mixture of aqueous ammonia solution (NH_4_OH, 0.5 mL for 320 nm-diameter mNPs), ethanol (40 mL), and deionized water (90 mL) was stirred at room temperature for 30 min. This was followed by addition of 0.5 g of dopamine hydrochloride dissolved in water. A gradual change in color of the solution from light brown to dark brown was observed. The reaction was continued for 24 h, and the formed mNPs were extracted through centrifugation at 7000 rpm, and washed three times with deionized water. Different sizes of nominally spherical nanoparticles were obtained by varying the volume of ammonium hydroxide, while following the same protocol.

### 2.3. Preparation of Glucose-Coated Melanin Nanoparticles, mNP@G

As-prepared mNPs (20 mg) were dissolved in tris-buffer (0.01 M, pH 7.5) and stirred for 10 min followed by addition of 0.5 g acetylglucosamine sugar dissolved in 10 mL deionized water. The reaction mixture was stirred for 24 h and then collected through centrifugation. After repeated washing with deionized water, the obtained mNP@G product was redispersed in 1.5 mL of deionized water for further characterization.

### 2.4. Cell Viability Measurements

Using Cell Counting Kit-8 assay (CCK-8, Sigma-Aldrich, St. Louis, MO, USA), cell viability was determined according to the manufacturer’s protocol with some modification, as explained in Appendix A. The CCK-8 colorimetric assay involves metabolic bioreduction of WST-8 [2-(2-methoxy-4-nitrophenyl)-3-(4-nitrophenyl)-5-(2,4disulfophenyl)-2H-tetrazolium, mono-sodium salt] in the presence of 1-methoxy PMS as an electron mediator, producing a water-soluble orange colored formazan dye. The amount of produced formazan is directly proportional to the number of living cells and can be measured by spectrophotometric method via absorbance of 460 nm. For these measurements, VM-M3, A375, and HeLa cancer cells were seeded, and injected at controlled concentrations into 24 well plates. The cells were incubated overnight in 5% CO_2_ atmosphere at 37 °C to allow adherence to the plate. For comparison, the cells were treated with different concentrations of mNPs and also mNP@G, and the plates were returned to the incubator for 15 h. Afterwards, 50 µL of CCK-8 was added to every well in culture medium, and after incubation for 3 h, the upper orange solution was removed, collected in centrifuge tubes, and centrifuged for 15 min at 10,000 rpm in order to remove the melanin nanoparticles from the formazan solution. Thereafter, the absorbance of the solution from every tube was measured separately by spectrophotometer (Lambda 950, PerkinElmer, Boston, MA, USA) in the wavelength range of 350–550 nm, and the viability was calculated at the maximum absorption wavelength (460 nm).

### 2.5. Biocompatibility and Cytotoxicity Measurements

Biocompatibility and cytotoxicity of various concentrations of mNPs and mNP@G from 140 to 2100 µg·mL^−1^ were studied via cell viability and proliferation of the A375 and HeLa cell lines, with the latter also studied in high (4500 mg·L^−1^) and low glucose (1000 mg·L^−1^) growth media, using CCK-8 assays kits (Sigma-Aldrich, St. Louis, MO, USA). To complete this study, we used UV–Vis spectrophotometry to evaluate cell viability, cell membrane damage and cell toxicity.

### 2.6. Theoretical Estimate of the Thermal Effects

To estimate the photothermal response of the in-blood circulating tumor cells sensitized with mNPs, we modeled the cell as a water droplet with average radius *r_c_* ≈ 5 × 10^−6^ m, immersed in blood serum, which for simplicity is also modeled as water, with thermal conductivity *k_m_* = 0.6 W·K^−1^·m^−1^, specific heat *c_w_* = 4186 J·kg^−1^·K^−1^, and density *ρ_w_* = 1000 kg·m^−3^. The initial temperature of the cell and the blood is *T_0_*. The cell is filled with a number *N* of much smaller, but highly radiation-absorbing mNPs. Each mNP has radius of *r_m_*≈10^−7^ m and thermal conductivity *k_m_* = 0.1 W·K^−1^·m^−1^. The time evolution of the average temperature change Δ*T_c_* = *Tc − T0* of a single cell is approximately given by:(1)$$\propto \frac{\partial \Delta T_{c}}{\partial t}\approx \emptyset -\beta \Delta T_{c}$$
where $\propto =\frac{4}{3}r_{c}\rho_{w}c_{w}$, $\beta =4k_{w}/r_{c}$, and ∅ is the radiation power density absorbed in all mNPs, given approximately by $\emptyset \approx \emptyset_{inc}N{\left(\frac{r_{m}}{r_{c}}\right)}^{2}$, under the assumptions that the mNPs absorb the radiation perfectly and the radius of the laser beam in our experiment is roughly $r_{c}$. The solution to Equation (1) is
(2)$$\Delta T_{c}=\left(\emptyset /\beta \right)\left[1-\exp\left(-\frac{\beta }{\propto }t\right)\right]$$

The maximum temperature increase is achieved for $\frac{\beta }{\propto }t\gg 1$, or for $t\gg \frac{\propto }{\beta }=\frac{r_{c}^{2}\rho_{w}c_{w}}{3k_{w}}=t_{c}$, such that $\Delta T_{cmax}=\emptyset /\beta =\emptyset r_{c}/4k_{w}$. Choosing a value of *N* = 1000, which corresponds to a low mNP load, and a power density as applied in this work, $\emptyset_{inc}\approx \;10^{9}$ W·m^−2^, we estimate ∅≈ $4\times 10^{10}$ W·m^−2^. With $\beta =0.5\times 10^{6}$ W·Km^−2^, Equation (2) gives $\Delta T_{cmax}\approx 1000\;K$. This is the order of heating that causes rapid boiling of the cell interior, and can lead to the cell lysis observed in our experiment, as discussed later. The time to achieve such a level of heating is of the order of $t_{max}\gg t_{c}\approx 10^{-4}$ s. It is important to note that this power density has no effect on the cells not having mNPs and immersed in water, since the penetration length in water at this frequency is very large, η≈10 m. Thus, a negligible fraction of the incoming radiation, of order ∅≈ $\emptyset_{inc}\frac{2r_{c}}{\eta }\approx \emptyset_{inc}10^{-6}=10^{3}$ W·m^−2^, is absorbed in a cell, in general agreement with our experiments. It is also in good agreement with laser tweezer experiments [38] in which cells, free of any nanoparticles, were subjected to NIR radiation with power density $\emptyset_{inc}=3.8\times 10^{10}$ W·m^−2^ for 15 min. It was shown in that experiment that this much larger power density and dose as compared with our experiment caused no delay in cell growth or increased mortality. Our simple estimate thus well explains the basic physics of our experiments with radiation.

## 3. Results and Discussion

### 3.1. Characterization of mNP@G

Figure 1 shows SEM images of as-prepared mNPs of different sizes (between 100 and 300 nm, ± 17 nm), obtained by variation of the NH_4_OH solution volume. As seen in Figure 1f, the mNP diameter is a linear function of the solution pH. As shown in Figure 2, a slight increase in size, e.g., from 145 nm for mNPs to 166 nm for mNPs@G (i.e., with ~10 nm average glucose coating thickness), was observed after surface functionalization with amino sugar, indicating surface coverage by the glucose. Figure 2c shows the optical absorption of aqueous solutions containing mNPs and mNP@G at the same concentration (0.1 g·L^−1^), recorded using UV–Vis-NIR spectroscopy. The spectra are similar, with the higher absorption of mNP@G in the visible range due to glucose coating. Surface functional groups of nanomaterials intended for biomedical application are crucial for their hydrophilicity and dispersibility in water and various biofluids. Thus, the chemical groups of melanin and the corresponding sugar-coated analog samples were determined using FTIR spectroscopy (Figure 2d). The intense C=O stretches from aromatic rings and/or carboxyl groups of the mNPs that almost suppressed other peaks can be observed at 1685 cm^−1^. The broad OH stretch of glucosamine alone can be visibly seen between 2700–3500 cm^−1^. The FTIR of the mNPs@G displayed an overlap of NH_2_/OH stretching around 3200–3500 cm^−1^. The C–O and C–C vibrational band arising from Schiff’s base reaction can be seen at 1097 cm^−1^ and 1298 cm^−1^ in the mNPs@G (Figure 2d), indicating successful functionalization.

### 3.2. Cell Viability after mNP and mNP@G Uptake

The main observations in this part of our study are: (1) non-cancerous or non-malignant cancer cells studied here do not ingest mNPs, (2) the studied malignant cancer cells massively absorb mNPs (macrophagic/phagocytic character), (3) this uptake is much stronger for the glucose-coated mNPs, (4) cell viability diminishes with increasing number of absorbed mNPs, and (5) lower glucose content in the cell nutrition dramatically reduces cell viability.

Figure 3 exemplifies the observations (1) and (2). It shows optical microscope images of a VM-M3 cell (top panels) and a VM-NM1 cell (bottom panels), both exposed to approximately the same amount of mNPs (26 h incubation time), and taken with focal planes at increasing depth into a cell (from left to right). This allows one to view cell interiors, and the nominal location of the absorbed nanoparticles. Clearly, the malignant VM-M3 cell contains mNPs throughout its interior. In contrast, the non-malignant VM-NM1 cell has no mNPs in its interior; these agglomerated in large clumps outside the cell. This confirms that only the malignant cells have phagocytic behavior. Similarly, in several similar tests (not shown here), we observed that healthy, non-cancerous cells also do not ingest mNPs.

Figure 4 exemplifies observations (3), (4) and (5) listed above. Figure 5a shows A375 cell viability versus concentration of mNP@G for two different glucose concentrations in the growth medium, and Figure 3b shows similar effects for the HeLa cells that normally, in contrast to the melanoma cells, contain no melanin nanoparticles. For more details, see the Appendix A. Note that a much longer incubation time (62 h) was used for higher concentrations of glucose in the growth medium compared to lower concentrations. This is simply because cells absorb the molecular glucose from the growth medium before they begin absorbing the much larger, glucose-coated nanoparticles. Thus, the incubation time approximately scales with the glucose concentration in the growth medium. The reason for higher cell viability at the same mNP concentration in high glucose concentration is not entirely clear. However, it is consistent with the Warburg effect according to which cancer cells benefit from increased amounts of glucose in the medium. One mechanism could be strengthening the cancer cell metabolism, which reduces the cytotoxic effects of mNPs. These suggest that a low glucose diet (e.g., ketogenic) for cancer therapies based on nanoparticles could be beneficial. Our results are in broad agreement with other reports on a variety of cancer cells [47,48,49,50].

The mechanism of the mNPs cytotoxicity is unclear at this point. It might be due to the nanoscopic size of the mNPs, which dramatically increases surface area for molecular chemical reactions with the cell interior components. Note that the melanin produced by melanocytes occurs in the form of microcrystals (average diameter *D*), much larger than mNPs in the current work (each with average diameter *d*) and thus, for the same melanin volume, have much smaller surface area (approximately *d/D*). If melanin had some finite surface-based cytotoxic effect, it would be expected to be enhanced with mNPs. Biologically active melanin has indeed been reported to be cytotoxic [51].

### 3.3. Cell Viability after Exposure to Radiation

Our custom-designed laser system employed a 532 nm wavelength diode pumped solid state laser, coupled to the input port of a fluorescent microscope. The beam was aligned and centered to the back aperture of an objective, and reflected light was filtered with a dichroic mirror. The sample was viewed and data recorded via Thorcam. The laser spot size on the sample was determined by the knife edge technique. As the blade moves across the laser spot, the measured laser light power *P* varies from zero to *P_max_*, and the shortest distance between these corresponding edge locations is recorded. This measurement is averaged over different heights and the diameter of the spot is then extrapolated by fitting a hyperbolic equation. Figure 5a shows a scaled plot of *P* vs. position *x* (with arbitrary origin) and the insets sketch approximate blade-beam locations at selected points. Figure 5b shows an optical image of the laser spot (with diameter *D* ≈ 7 μm, marked by a yellow circle) on a fluorescent card, for a chosen magnification setup on the microscope. 

Figure 5c–e shows images of two VM-M3 cells moderately filled with mNPs, at various exposure times to laser radiation, and at an identical magnification setup. Due to filtering, the laser spot is invisible, so its outline is marked with a ~50 μm diameter yellow circle. The power density is moderate, ∅*_inc_* ≈ 6 × 10^7^ W·m^−2^, but enough to initiate visible cell damage after 15 s of exposure, and catastrophic cell damage (explosion) after 30 s. To better match the size of the laser spot to cell dimensions, we have changed the magnification setup and laser power. The effect of that scenario on a (different) VM-M3 cell, again moderately filled with mNPs, is shown in Figure 5f–h. This time, the laser spot (also marked with a yellow circle) has diameter ~7 μm, and the corresponding power density is much larger, ∅*_inc_* ≈1.4 × 10^9^ W·m^−2^. The figure shows that the damage is now very localized, clearly starting at the clusters of mNPs, with damage obvious already after 0.1 s exposure, and complete catastrophic cell damage after only 3 s. The white spots visible inside the high laser intensity regions are due to photoluminescence of highly excited mNPs. The results in this figure confirm a well-known fact that visible light can inflict damage to cells, including catastrophic damage. Such a process would be of little therapeutic use if it was not selective. 

We found that the level of radiation capable of catastrophically destroying mNP-filled VM-M3 cells, like in Figure 5, is safe for VM-M3 cells not sensitized with mNPs. In this experiment, mNP-unfilled cells were grown on a microscopic slide, marked with a 1 mm × 1 mm grid to track cells throughout the experiment. The optical microscope image of the slide, shown Figure 6a, was taken 24 h after exposure to laser light of cells in the box numbered 5 (solid-red outline). The cells experienced the same power density (and approximately the same exposure time) as the cell shown in Figure 5h. The cells in the bracketed boxes 1–4 were not exposed to the laser. During the 24 h, the cells were incubated, and at the end, the Live/Dead Cell Staining Kit II (PromoKine) assay was applied, to visualize cell viability. Figure 6a shows that the cells take up the Calcein-AM dye, resulting in green fluorescence, and are not permeable to the EthD-II dye, which would result in red fluorescence. Thus, all cells remain alive.

Figure 6b shows a population bar diagram in the corresponding 5 boxes, where each box is represented by two bars: red bars before and blue bars after laser illumination of box 5. The red bars show that the initial cell distribution was roughly uniform in boxes 2–4 (with an average cell number per box ~47), box 1 had ~25, and box 5, ~75 cells. After 24 h, the number of alive cells increased in all boxes (growing cells), including laser-exposed box 5. It is also clear that the overall distribution of cells on the grid changed, with the number of cells per box gradually increasing, e.g., to ~50 in box 1 and to ~90 in box 5. This effect likely results from a combination of natural cell population growth and temperature rise from laser heating of box 5, and the heat transfer away from this box. The resulting temperature profile during illumination would then be asymmetric, with a gradual temperature drop towards box 1, with cell growth reflecting this profile. Recorded movies of the details of these mNP-filled cells explosively damaged, as well as of the unfilled cells remaining unaffected, can be found in the Appendix A. To demonstrate the lack of visible damage at the microscopic level, we show optical images of a single, mNP-unfilled MV-M3 cell before (panel c), and after (panel d) laser illumination, at the same level as applied in panel (a). As expected, the radiation produces no visible change in the cell.

The main results in this part of our study are: (a) compared with the cytotoxicity of nanoparticles alone, laser-induced cell death requires much lower density of absorbed nanoparticles, (b) all cells filled with absorbing nanoparticles (e.g., mNPs) are destroyed by radiation, at sufficient power level—this would include melanoma cells, naturally filled with melanin microcrystals, and (c) there is a laser power range at which the nanoparticle-filled cancer cells are violently destroyed, while the nanoparticle-free cells remain alive. This is a key finding of this work, since the nanoparticle-filled cells do not have to be so violently destroyed to be killed, and so the applied laser power level can be strongly reduced. This lower power level obviously will not damage the nanoparticle-free cells.

Our nanoparticle-based strategy could be used as a basis for or part of a cancer therapy (e.g., optochemotherapy), for example to target circulating tumor cells which mediate metastasis. In such a therapy, an intravenous injection could accomplish the first stage of the mNPs feeding into CTCs. This step could be enhanced by additional bio-engineered CTC targeting schemes. Next, in one possible scenario, one could expose the blood of a cancer patient to light externally, in a dialysis-like scheme. This would lead to a dramatic reduction in the CTC population, thus significantly reducing the effects of metastasis. 

## 4. Conclusions

We have observed massive cellular uptake of melanin nanoparticles by the studied metastatic cancer cells (macrophagic/phagocytic character) which, at sufficiently high density, causes a cytotoxic effect. This effect is further enhanced by coating the nanoparticles with glucose, and simultaneous reduction of the glucose level in the growth medium. We also demonstrated that nonionizing visible light at moderate power levels kills these metastatic cancer cells, at much lower mNP uptake levels. Cell death occurs in this case via hyperthermia-induced lysis, and we found this process to be target-selective, as nonmalignant cancer cells studied here that could not ingest the melanin nanoparticles remain unaffected, despite receiving identical optical energy levels and doses. This technique could enhance a future cancer metastasis preventing therapy.

## Figures and Tables

**Figure 1 pharmaceutics-13-00965-f001:**
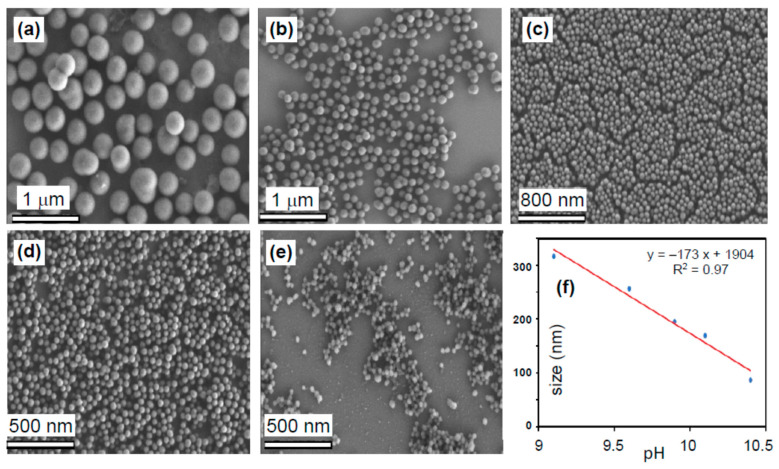
SEM images of mNPs in different volumes of NH_4_OH: (**a**) 0.5 (**b**) 1.0, (**c**) 1.5, (**d**) 2.0, and (**e**) 2.5 mL. (**f**) Plot of mNP diameter vs. pH of NH_4_OH solution.

**Figure 2 pharmaceutics-13-00965-f002:**
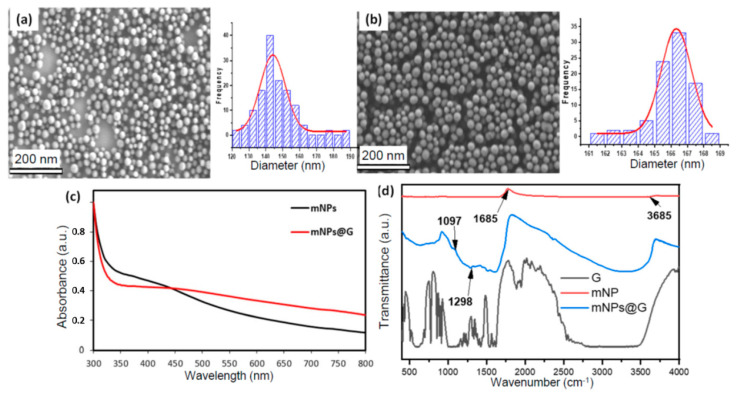
(**a**,**b**) SEM images and the corresponding size distribution histograms for mNP and mNP@G, respectively; (**c**) UV-Vis absorption of mNP (black) and mNP@G (red line). (**d**) FTIR spectra of terminal amino glucose (black line), mNP (red line), and mNPs@G (blue line).

**Figure 3 pharmaceutics-13-00965-f003:**
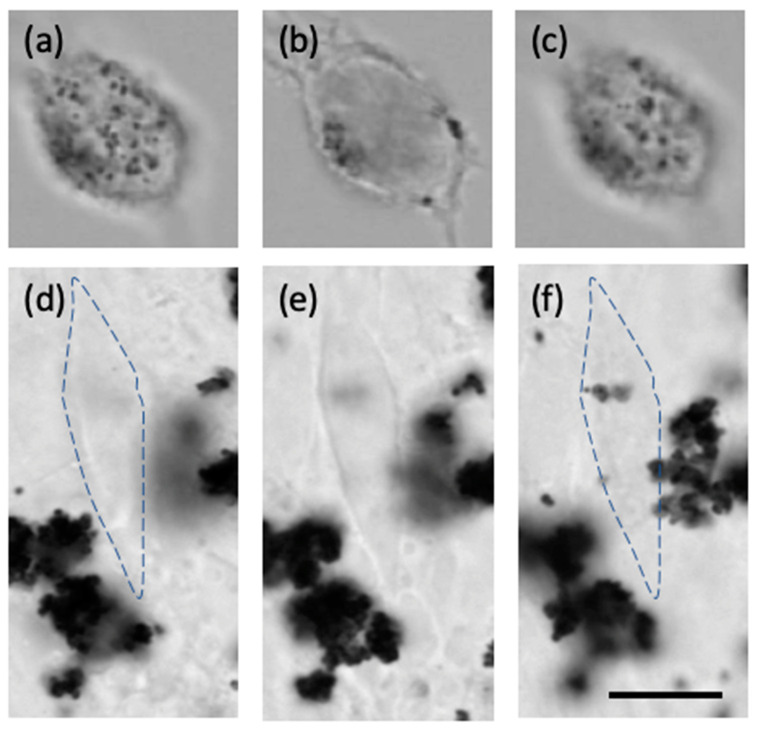
Optical microscope images of a VM-M3 cell (**a**–**c**) and a VM-NM1 cell (**d**–**f**), taken at changing focal plane depths: near the top cell surface (**a**,**d**), middle of the cell (**b**,**e**), and near the bottom cell surface (**c**,**f**). Both cells are shown after 26 h incubation with mNPs. The common scale bar is 10 m. The dashed line in (**d**,**f**) outlines the cell shape as in (**e**).

**Figure 4 pharmaceutics-13-00965-f004:**
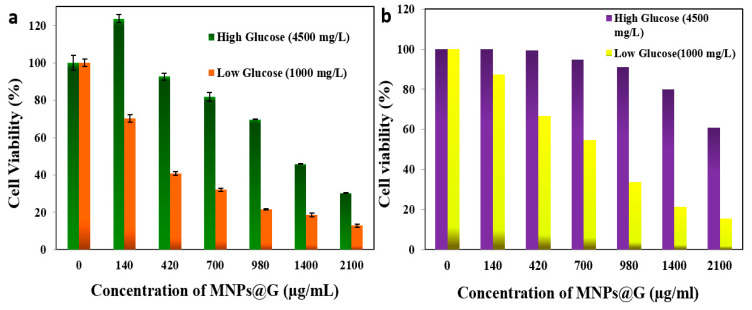
(**a**) A375 melanoma and (**b**) HeLa cell viability according to CCK-8 assay as a function of mNP@G concentration after 62 h incubation in high glucose growth medium (4500 mg·L^−1^) and 15 h incubation in low glucose growth medium (1000 mg·L^−1^).

**Figure 5 pharmaceutics-13-00965-f005:**
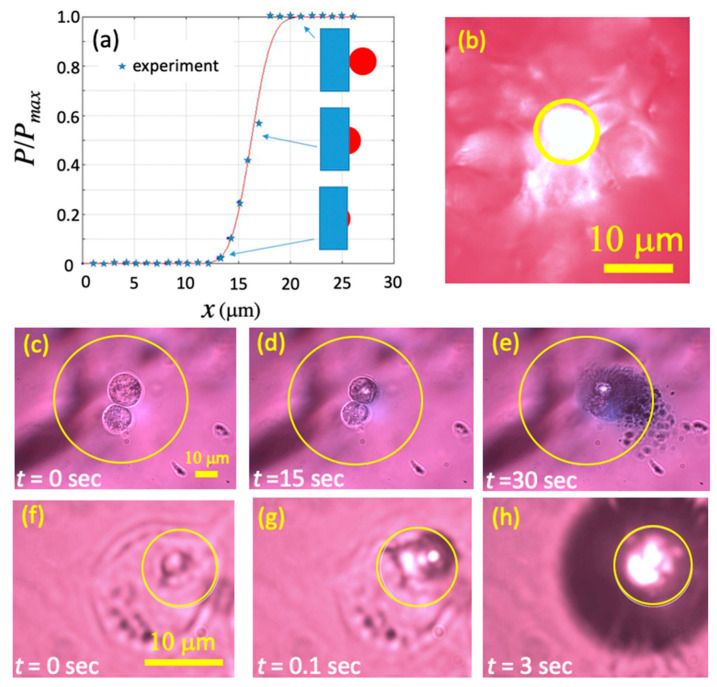
(**a**) Optical power (scaled to its maximum) versus position, measured using the blade edge shading effect. The insets sketch approximate blade-beam relative locations at selected points. (**b**) Optical image of the laser spot obtained by using the fluorescent card. (**c**–**e**) Images of two VM-M3 cells moderately filled with mNPs, at various exposure times to laser light, at 10× magnification (light power density ∅_i*nc*_ ≈ 6 × 10^7^ W·m^−2^). (**f**–**h**) Images of a VM-M3 cell moderately filled with mNPs, at various exposure times to laser radiation, at 100× magnification (power density ∅*_inc_* ≈ 1.4 × 10^9^ W·m^−2^). Yellow circles mark approximate beam diameter, which changes with magnification.

**Figure 6 pharmaceutics-13-00965-f006:**
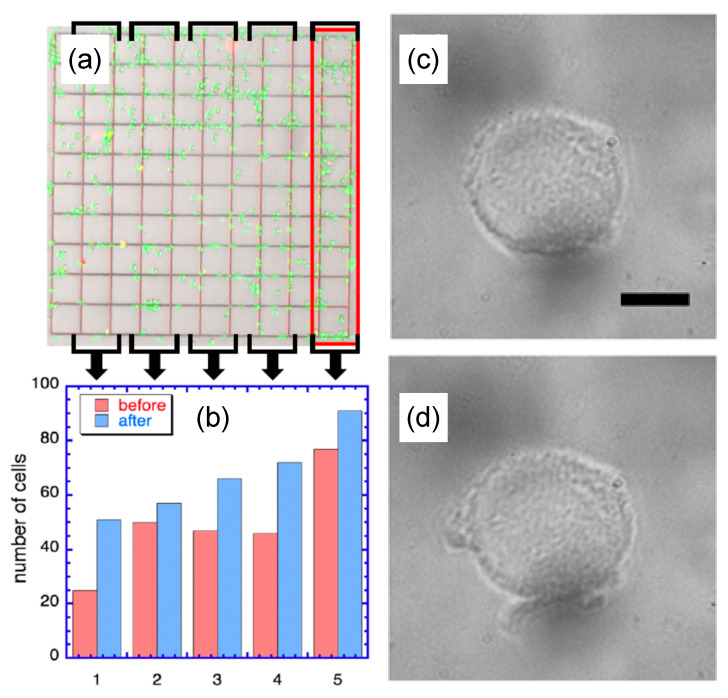
Effect of laser light on VM-M3 cells unfilled with mNPs. (**a**) Live/dead cell staining test: all cells luminesce green and none red, indicating all are alive and growing. Only cells within the red-outlined, bracketed box have been exposed to the laser. (**b**) Bar diagram of cell counts in boxes 1–5, with red = before and blue = 24 h after laser exposure of cells in box 5, all in growth medium. (**c**) An mNP-free, single VM-M3 cell before and (**d**) after laser exposure as described in the text. Scale bar 10 µm.

## Data Availability

The data presented in this study are available on request from the corresponding author.

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
