# Peer review of "Light- and Melanin Nanoparticle-Induced Cytotoxicity in Metastatic Cancer Cells"

_pharmaceutics, 2021, doi:10.3390/pharmaceutics13070965_

Round 1
Reviewer 1 Report
The quality of manuscript seems to be improved compared to previous version.
Author Response
We thank the Reviewer for the positive feedback.
Reviewer 2 Report
Although I appreciate the efforts of the Authors to improve their manuscript, several major concerns still arise, and I feel that the manuscript should be considered for publication following major revisions.
In particular, the Authors substantially ignored all the most important questions that I raised. In the revised version of the manuscript, it is still not clear what is the mechanism through which a high concentration of glucose-coated melanin NPs exerts a cytotoxic effect in cancer cells in the absence of radiation. Are these nanoparticles toxic per se? Why, as they are made only of melanin and glucose (which, obviously, are not toxic)? Do these nanoparticles interfere with some important biological process? This is a point that must be discussed in the manuscript since the nanoparticles could be further functionalized to improve their uptake by CTCs and achieve an anticancer effect even without the need of non-ionizing radiation.
The second fundamental point that the Authors completely ignored is the real selectivity of this approach. At lines 369-370 the Authors state that “healthy cells that could not ingest the melanin nanoparticles remain unaffected, despite receiving identical optical energy levels and doses”. However, the Authors never performed an experiment to demonstrate that healthy cells do not ingest the melanin nanoparticles and survive to non-ionizing radiation with the same energy levels and doses used to kill mNP-filled cancer cells. They only demonstrated that VM-M3 cells (which do not contain endogenous melanin) without mNPs survive to the same doses of non-ionizing radiation with the same energy levels that kills mNP-filled VM-M3 cells. This is clearly not sufficient to state that healthy cells are spared by these nanoparticles unless the Authors demonstrate it with an experiment. Since they suggest using these nanoparticles to kill CTCs, they must demonstrate that leukocytes (in particular of the myeloid lineage) do not phagocytize mNPs. Without this demonstration the Authors cannot state that such nanoparticles are not internalized by healthy cells.
Third point that has been ignored by the Authors: why all the experiments in which non-ionizing radiation is used have been performed on a histiocytoma cell line (which do not contain melanin) rather than on melanoma cells, which are highly pigmented with melanin? As I asked in my previous report, why melanoma cells, which are already filled with melanin, are not destroyed by a laser beam? At lines 311-312 the Authors state that “the level of radiation capable of catastrophically destroying mNP-filled VM-M3 cells, like in Figure 5, is safe for VM-M3 cells not sensitized with mNPs”. Visible light is safe for VM-M3 cells without mNPs only because they do not contain melanin. What about melanin-rich melanoma cells?
Other points:
Title: the cytotoxicity of light and melanin nanoparticles is exerted in metastatic cancer cells. Thus, the title should be amended as follow: "Light- and melanin nanoparticle-induced cytotoxicity in metastatic cancer cells", rather than "... of metastatic cancer cells".
Keywords: I do not understand the utility of the keyword "Formazan".
Line 94: melanin affects melanoma behavior.
Line 143: dissolve should be substituted with dissolved.
Lines 170-171: I do not understand why the Authors indicate here the diameter of mNPs and of mNP@G.
Line 223: it is not Figure 3d, it is Figure 2d.
Line 231: please correct NH4OH.
Section 3.2: I am convinced that in high glucose media cancer cells require more time to internalize glucose-coated NPs. However, if "incubation time approximately scales with the glucose concentration in the growth medium", why is the cytotoxic effect different between high glucose and low glucose conditions? I thought that the Authors chose these time points to achieve a comparable intake of mNP@G in cancer cells, thus the cytotoxicity should be also comparable. If not, I do not understand why the Authors chose these time points. What do they want to demonstrate? That cancer cell killing is much more efficient in a low glucose regimen as cancer cells absorb free glucose better than glucose-coated NPs? Then, the Authors must show the effect of high glucose at the same time point of low glucose condition. It makes no sense to show two different time points that are in no way comparable (I mean, based on the results shown by the Authors, 62 hours of incubation in high glucose medium are not sufficient to induce an internalization of the NPs that is comparable to the internalization achieved following 15 hours of incubation in low glucose medium, so why did they choose 62 hours and not more?).
Line 322: please correct "Calcien" with "Calcein".
Line 331: "This effect likely results as a combination...".
Lines 331-335: comparing the increase in cell number for all the segments, it does not seem to me that cell growth reflects the thermal profile: the increase in cell number after the exposure of segment 5 to light is equal in segments 1 and 4, while it is minimum in segment 2.
Figure 6: (a) and (b) are not indicated in the figure.
Lines 364-365: this sentence is misleading. Coating the NPs with glucose exploits the Warburg effect as cancer cells use glycolysis even when oxygen is abundant, so they uptake much more glucose than healthy cells. The Authors should explain this rather than simply writing "Warburg effect" in brackets.
Line 371: as this strategy aims at drastically reducing the number of CTC, it is inappropriate to state that this approach could be the basis or part of a cancer therapy targeting metastasis, as it would rather prevent the onset of metastases.
I agree with the Authors that killing all CTCs is impossible and not necessary to efficiently prevent the onset of metastases. However, I still believe that the strategy proposed by the Authors is improbable as geometrically targeting isolated cells using a laser beam with a diameter of less than 100 µm in a dialysis-like scheme looks like finding a needle in a haystack. It is true that it is not necessary to induce the explosion of cancer cells to kill them, so the laser power could be reduced, but I am not convinced about the possibility to irradiate all the blood of a patient with a sufficient energy to induce the death of CTCs.
Author Response
Response to Reviewer 2
We thank the reviewer for again a very thorough review of our revised work. We want to stress that we appreciate that effort, and sincerely feel that working on these issues has improved our paper. Before responding in detail, let us summarize the main points of our work. We have completely revised the Abstract (lines 25-36) and Conclusions (lines 417-426) of our revised manuscript to accommodate these points. Note, that for the reviewer convenience, we marked (temporarily) all changes red in the revised manuscript.
This work is an ongoing endeavor, involving researchers from 4 disciplines: biology, chemistry, physics and materials science. The presentational difficulties may follow from that interdisciplinary culture. The work, at this point, is admittedly far from complete. Our main observations reported in this manuscript are:
- Metastatic VM-M3 cancer cells studied here, which have characteristics of macrophages, massively ingest melanin nanoparticles (mNPs).
- The observed ingestion is enhanced by coating mNPs with glucose.
- After a certain level of mNP ingestion, the metastatic cancer cells studied here are observed to die; glucose coating appears to slow that process. The cell death mechanism is not clear at the moment. We will later seek to determine if an apoptosis mechanism can account for the cell death.
- Cells that accumulate mNPs are more susceptible to killing by laser illumination than cells that do not accumulate mNPs.
- Non-metastatic VM-NM1 cancer cells studied in his work do not ingest the mNPs, and remain unaffected after receiving identical optical energy levels and doses.
Below are our answers to the specific questions of this reviewer, embedded in the reviewer comments.
Reviewer: In the revised version of the manuscript, it is still not clear what is the mechanism through which a high concentration of glucose-coated melanin NPs exerts a cytotoxic effect in cancer cells in the absence of radiation. Are these nanoparticles toxic per se? Why, as they are made only of melanin and glucose (which, obviously, are not toxic)? Do these nanoparticles interfere with some important biological process? This is a point that must be discussed in the manuscript since the nanoparticles could be further functionalized to improve their uptake by CTCs and achieve an anticancer effect even without the need of non-ionizing radiation.
Answer: While we feel that the cytotoxicity of mNPs is a significant observation, worthy of publication, we acknowledge that the reasons for this effect are unclear, and this requires further work. It might be due to the nanoscopic size of the mNPs, which dramatically increases surface area for molecular chemical reactions with the cell interior components. Note that the melanin produced by melanocytes occurs in the form of microcrystals, much larger than mNPs in the current work, and thus, for the same melanin volume, have much smaller surface area. So, if melanin has some finite, surface-related cytotoxic effect, it could be expected to be enhanced with mNPs. We observed that glucose coating seems to prevent cytotoxicity, and that the cells begin to die rapidly only after the very thin glucose coating on mNPs is “eaten up” by the cell. Also, the nanoparticles may not necessarily be toxic per se; biologically-active melanin has indeed been reported to be cytotoxic (see http://dx.doi.org/10.1155/2014/306895).
We now made this point in the revised version of the manuscript (line 294-301). We have also added the new reference [51].
Reviewer: The second fundamental point that the Authors completely ignored is the real selectivity of this approach. At lines 369-370 the Authors state that “healthy cells that could not ingest the melanin nanoparticles remain unaffected, despite receiving identical optical energy levels and doses”. However, the Authors never performed an experiment to demonstrate that healthy cells do not ingest the melanin nanoparticles and survive to non-ionizing radiation with the same energy levels and doses used to kill mNP-filled cancer cells. They only demonstrated that VM-M3 cells (which do not contain endogenous melanin) without mNPs survive to the same doses of non-ionizing radiation with the same energy levels that kills mNP-filled VM-M3 cells. This is clearly not sufficient to state that healthy cells are spared by these nanoparticles unless the Authors demonstrate it with an experiment.
Answer: The reviewer is correct. It was our mistake to write that “healthy cells that could not ingest the melanin nanoparticles remain unaffected, despite receiving identical optical energy levels and doses.” What we did show is that malignant cancer cells without the melanin nanoparticles, or nonmalignant cancer cells that could not ingest the melanin nanoparticles remain unaffected, despite receiving identical optical energy levels and doses (our point v above).
We now removed the incorrect statements (including the former lines 369-370) from the revised version of the manuscript. We replaced those with correct statements in the Abstract and in the Conclusion.
We have also modified Figure 3 by replacing the bottom panels with z-stack images of the non-malignant VM-NM1 cell exposed to mNPs. In spite the exposure, this cell contains no mNPs in the interior (no ingestion).
Additional comments:
The effect of laser cell killing is physically well understood. It is due to mNPs strongly absorbing light at the chosen frequency, and thus overheating the cell interior. The relevant point here is the presence or absence of mNPs inside a cell. The cells unfilled with mNPs absorb orders of magnitude less light and remain unaffected, regardless of cell type tested. In this context, it was fully acceptable for us to choose the filled and unfilled VM-M3 cells.
Nevertheless, to further illustrate the effect, we have also modified Figure 6 by adding two panels with optical images of a VM-M3 cell (with no mNPs inside), and subjected to moderate non-ionizing radiation. As expected, the radiation caused no visible changes. Corresponding new text starts on line 365.
Reviewer: Since they suggest using these nanoparticles to kill CTCs, they must demonstrate that leukocytes (in particular of the myeloid lineage) do not phagocytize mNPs. Without this demonstration the Authors cannot state that such nanoparticles are not internalized by healthy cells.
Answer: It has been observed by others that other cells in the blood (besides macrophages) ingest nanoparticles at negligible levels compared to malignant cancer cells. However, we agree with the reviewer, that we cannot state in general that nanoparticles are not internalized by healthy cells. Therefore, we eliminated all such statements from the revised manuscript.
Reviewer: Third point that has been ignored by the Authors: why all the experiments in which non-ionizing radiation is used have been performed on a histiocytoma cell line (which do not contain melanin) rather than on melanoma cells, which are highly pigmented with melanin? As I asked in my previous report, why melanoma cells, which are already filled with melanin, are not destroyed by a laser beam? At lines 311-312 the Authors state that “the level of radiation capable of catastrophically destroying mNP-filled VM-M3 cells, like in Figure 5, is safe for VM-M3 cells not sensitized with mNPs”. Visible light is safe for VM-M3 cells without mNPs only because they do not contain melanin. What about melanin-rich melanoma cells?
Answer: We never meant to imply that melanoma cells “are not destroyed by a laser beam”. In fact, these can be (like any other mNP-filled cell) easily killed by laser illumination. This is an obvious physical effect, and we discuss it and provide a theoretical estimate in the Materials and Methods section of our manuscript. It has been demonstrated in Refs. [23] and [25] (see lines 79-82 of our manuscript) that melanoma CTCs, naturally filled with melanin nano/micro crystals, absorb an order of magnitude more light (in the spectral windows near 500 and 700 nm wavelength) than other cells in the blood which are not filled with melanin.
To make this point very clear, we have added an explanatory sentence on lines 403.
Additional comments:
The original idea of this work indeed started with melanoma cells. However, in order to generalize this concept, we focused on turning any cancer cell into a light-absorbing cell like melanoma, by filling it with engineered melanoma nanoparticles of well controlled sizes, shape and coating. The macrophage and the Warburg effects were to be the mechanisms to fill CTCs with the mNPs. Laser light was to make the final kill. However, during work on this idea, we discovered that even without laser exposure, the CTCs would die, after the mNP concentration becames sufficiently high.
Reviewer:
Other points:
Title: the cytotoxicity of light and melanin nanoparticles is exerted in metastatic cancer cells. Thus, the title should be amended as follow: "Light- and melanin nanoparticle-induced cytotoxicity in metastatic cancer cells", rather than "... of metastatic cancer cells".
- We agree, and have changed the title accordingly.
Keywords: I do not understand the utility of the keyword "Formazan".
- We have removed the word.
Line 94: melanin affects melanoma behavior.
- This has been corrected.
Line 143: dissolve should be substituted with dissolved.
- This has been corrected.
Lines 170-171: I do not understand why the Authors indicate here the diameter of mNPs and of mNP@G.
- This information was provided in support of the fact that the mNPs were indeed coated with glucose. We have added on line 218 “… from 145 nm for mNPs to 166 nm for mNPs@G (166 nm, i.e. with ~10 nm average glucose coating thickness)…”
Line 223: it is not Figure 3d, it is Figure 2d.
- This has been corrected.
Line 231: please correct NH4OH.
- This has been corrected.
Section 3.2: I am convinced that in high glucose media cancer cells require more time to internalize glucose-coated NPs. However, if "incubation time approximately scales with the glucose concentration in the growth medium", why is the cytotoxic effect different between high glucose and low glucose conditions? I thought that the Authors chose these time points to achieve a comparable intake of mNP@G in cancer cells, thus the cytotoxicity should be also comparable. If not, I do not understand why the Authors chose these time points. What do they want to demonstrate? That cancer cell killing is much more efficient in a low glucose regimen as cancer cells absorb free glucose better than glucose-coated NPs? Then, the Authors must show the effect of high glucose at the same time point of low glucose condition. It makes no sense to show two different time points that are in no way comparable (I mean, based on the results shown by the Authors, 62 hours of incubation in high glucose medium are not sufficient to induce an internalization of the NPs that is comparable to the internalization achieved following 15 hours of incubation in low glucose medium, so why did they choose 62 hours and not more?).
Answer: Yes, we chose different times to achieve a comparable intake of mNP@G in cancer cells. But no, the cytotoxicity does not have to be the same (comparable)! In a higher glucose background, cancer cells thrive since these are fermenters (Warburg effect), and thus survive the cytotoxicity on mNPs better (higher viability for the same mNPs concentration). However, the mechanism is not clear yet, and needs more studies.
- To clarify this, we added the following comment starting on line 282-287:
The reason for higher cell viability at the same mNP concentration in high glucose concentration is not entirely clear. However, it is consistent with the Warburg effect according to which cancer cells benefit from increased amount of glucose in the medium. One mechanism could be strengthening the cancer cell metabolism, which reduces the cytotoxic effects of mNPs. These suggests that a low glucose diet (e.g. ketogenic) for cancer therapies based on nanoparticles could be beneficial.
Line 322: please correct "Calcien" with "Calcein".
- This has been corrected.
Line 331: "This effect likely results as a combination...".
- We have changed to "This effect likely results froma combination...".
Lines 331-335: comparing the increase in cell number for all the segments, it does not seem to me that cell growth reflects the thermal profile: the increase in cell number after the exposure of segment 5 to light is equal in segments 1 and 4, while it is minimum in segment 2.
Answer: We disagree. After exposure to light there is a monotonic, overall higher temperature distribution away from segment 5.
Figure 6: (a) and (b) are not indicated in the figure.
- This has been corrected and two panels also added to Figure 6.
Lines 364-365: this sentence is misleading. Coating the NPs with glucose exploits the Warburg effect as cancer cells use glycolysis even when oxygen is abundant, so they uptake much more glucose than healthy cells. The Authors should explain this rather than simply writing "Warburg effect" in brackets.
- Agreed: “Warburg effect” in brackets removed
Line 371: as this strategy aims at drastically reducing the number of CTC, it is inappropriate to state that this approach could be the basis or part of a cancer therapy targeting metastasis, as it would rather prevent the onset of metastases.
- Agreed: statements changed.
I agree with the Authors that killing all CTCs is impossible and not necessary to efficiently prevent the onset of metastases. However, I still believe that the strategy proposed by the Authors is improbable as geometrically targeting isolated cells using a laser beam with a diameter of less than 100 µm in a dialysis-like scheme looks like finding a needle in a haystack. It is true that it is not necessary to induce the explosion of cancer cells to kill them, so the laser power could be reduced, but I am not convinced about the possibility to irradiate all the blood of a patient with a sufficient energy to induce the death of CTCs.
Answer: We disagree, in the sense that the laser power can be dramatically increased, and pulse and scanning schemes could be applied to expose large surface area of ultrathin fluidic channels of blood flowing through a the dialysis-like system. This could make this procedure practical (manageable exposure time per patient, manageable expense in a centralized treatment facility).
Round 2
Reviewer 2 Report
I really appreciate the efforts of the Authors to improve their manuscript. Although I am still not convinced about some points, the Authors replied exhaustively and in detail to all my observations. Thus, I now feel that their manuscript is worth of publication.
This manuscript is a resubmission of an earlier submission. The following is a list of the peer review reports and author responses from that submission.
Round 1
Reviewer 1 Report
I think that the topic of this study is somewhat interesting, however their hypothesis was not appropriately supported by robust scientific evidences. More elaborate experimental design is necessary for identifying their initial objective. I'm not able to recommend its acceptance after revision with current format.
1) In Figure 1, those data seem to be cited from other literatures. Can you make up it with your own experimental data?
2) If the glucose concentration is one of critical factors for cellular uptake, authors have to determine the glucose content in mNPs@G. More quantitative data should be provided except for UV-Vis and FT-IR data.
3) In Figure 4, did you measure the cell viability in no glucose condition? Why did you select melanoma and HeLa cells for CCK-8 assay? Why did you incubate cells 62 h for high glucose medium and 15 h for low glucose medium?
4) In Table 1, how did you prepare cell suspension for zeta potential analysis? Are there any reference zeta potential data?
5) In Figures 5-7, why did the authors use VM-M3 cells rather than other cells tested in Table 1? The resolution of those figures should be improved. If they want to verify the cellular uptake of mNPs or mNPs@G, more precise analytical methods are necessary to demonstrate them more clearly. For instance, cellular mNPs can be quantitatively analyzed by measuring the absorbances following cell lysis.
6) For verifying the hyperthermia-induced lysis for cancer cell killing, authors have to provide thermal image of mNPs (or mNPs@G) and direct evidences for cell lysis.
7) Cell death mechanisms including underlying molecular pathways should be elucidated.
Reviewer 2 Report
In the manuscript entitled “Nonionizing radiation and melanin nanoparticle cytotoxicity of self-sensitized metastatic cancer cells” Gabriele and colleagues synthetized glucose-coated melanin nanoparticles and characterized their anticancer activity against metastatic cancer cells when coupled with non-ionizing radiation. Although this topic could be of some interest from a therapeutic point of view, the study is marred by several unclear aspects and flaws and it is, thus, not suitable for publication. Several major concerns arise.
First, English language needs to be improved as the syntax is sometimes wrong and several typos are present. From a scientific point of view, it is not clear whether the Authors wanted to focus their paper on melanoma or, more generally, on metastatic cancers. In fact, the Introduction seems to focus the attention of the readers on melanoma, however the Authors used only a cell line derived from melanoma and most of the (scarce) results are relative to a histiocytoma cell line. Whether the Authors really wanted to describe a novel strategy aimed at killing the metastatic melanoma cells, the use of a single cell line derived from this type of tumor is a serious flaw of the study. In this regard, there is a terrible confusion about cell lines throughout the manuscript. At line 23 and at lines 113-114 (apart from the fact that "cancer cells melanoma" is wrong - it should be written "melanoma cancer cells" or “cancer cells originated from melanoma") the Authors wrote “cancer cells melanoma, HaCaT and HeLa”. From a grammatical point of view, this sentence means that HaCaT and HeLa cells are cancer cells of melanoma origin. I think it is not necessary to specify that HeLa, the most famous human cancer cells, were originated from the cervical carcinoma of Henrietta Lacks. HaCaT cells, instead – as specified by the Authors at line 127 – are spontaneously immortalized keratinocytes (so, technically, they are not metastatic cancer cells). Similarly, at line 126 the Authors wrote “Malignant melanoma A375 and HeLa cell lines were obtained…”. The meaning of this sentence is clearly that HeLa are malignant melanoma cells. I am confident that the Authors meant they used melanoma cancer cells and HaCaT and HeLa cells, but these sentences need to be rephrased as they are currently totally misleading! Moreover, as the only cancer cell line derived from melanoma used in this study is A375, I do not understand why the Authors wrote "HaCaT and HeLa" but did not indicate A375 (not only here, but throughout the whole manuscript; if they wrote HaCat and HeLa each time, why did they write melanoma instead of A375?).
One of the major critical point of the study is the cytotoxicity of glucose-coated melanin NPs in the absence of non-ionizing radiation. In the Introduction the Authors reported that melanoma cells are highly pigmented due to dysregulation of melanin synthesis. Melanin is not toxic, as its role is to protect skin from ultraviolet light. However, as melanin NPs are toxic and could be excited by non-ionizing radiation leading to hyperthermia, why are melanoma cells not susceptible to overheating when targeted by a laser beam, being highly pigmented with melanin? This point needs to be clarified. Moreover, it remains unclear why melanin NPs should be toxic to cells. As melanin is (obviously) not toxic, what is the mechanism of cytotoxicity induced by glucose-coated melanin NPs?
The Authors claim that glucose-coated melanin NPs are selective against cancer cells, as they are not ingested by healthy cells. However, the phagocytic behavior is not a peculiarity of cancer cells. Several normal cells – think about macrophages – are able to ingest extracellular materials. It is not necessary to remember that, in the context of skin pigmentation, keratinocytes phagocytize melanin-containing melanosomes originated from melanocytes. Thus, as they do not present data showing that glucose-coated melanin NPs are safe to healthy cells, how can the Authors state that this strategy is not toxic to epidermal cells?
Lines 45-46: pigmentation in melanoma cells is not unregulated, rather it is dysregulated. In fact, considering the role played by melanin in the fate of cancer cells, as hinted at by the Authors in the following lines, pigmentation needs to be tightly regulated during tumorigenesis.
Lines 121-122: why cancer cells that internalized melanin NPs are defined self-sensitized? This definition is wrong, as cells are sensitized by NPs, not by themselves. If so, all cancer cells that uptake drugs through active transport should be defined self-sensitized. Please also amend the title of the manuscript.
Line 157: "at an identified concentration". The Authors should report which one for every cell line used.
Lines 185-190: the explanation of CCK-8 colorimetric assay should be moved where it is reported for the first time (in paragraph 2.4). Why are two different absorption wavelengths (450 nm in paragraph 2.4 and 460 nm in paragraph 2.6) indicated for the same assay?
Line 266-268: the Authors should explain why cell viability in high vs low glucose growth medium is evaluated at two very different time points (62 h vs 15 h). Moreover, why cell viability is reduced by melanin NPs even in the absence of non-ionizing radiation?
Lines 273-274: as HeLa are probably the most transformed human cancer cells, why is the zeta potential of A375 more negative? Are surface electro-kinetic properties of these cells diverse?
Lines 279-284: it is not clear why the Authors chose to use an optical microscope (unable to focus different focal planes) to take pictures with focal planes at increasing depth. Obviously, the resulting pictures are all blurred. An optical microscope already shows the maximum projection of all focal planes composing an object. Moreover, why did the Authors choose to use a histiocytoma cell line instead of A375 cells, which - as reported by the Authors themselves at line 114 - have a "macrophage character"?
Figure 6a is not clear. Please explain better.
Figure 7: why are cells in Figure 7b and 7c round and shrinked compared to cells shown in Figure 7a? Live/Dead assay clearly shows that these cells are alive, however they really look like hematological cells rather than adherent cells (as those shown in Figure 7a). Moreover, Figure 7d, which should show a magnified optical image of a typical cell from the illuminated box, is completely blurred, so that it is impossible to understand the morphology of the cell (the morphology is clearer in movie S2 in the Supplementary Materials, however the cells shown in that movie seem to be highly different from those shown in Figure 7b and 7c). It seems that data reported in this Figure are false.
Why is a Discussion section missing? The Conclusions are too concise. The role of low glucose levels in growth medium is not discussed (it is obvious that, reducing glucose levels, cancer cells increase the uptake of glucose-coated NPs, but this should be discussed in the view of Warburg’s effect). Moreover, I think that at least the advantage of coating the melanin NPs with glucose should be discussed from a therapeutic perspective. Could these NPs be associated with a ketogenic diet to improve their anticancer efficacy in patients? From a therapeutic point of view, however, this strategy is meaningless. It is highly improbable that one could kill all CTCs circulating in the blood of a cancer patient by irradiating all the blood volume in a dialysis-like scheme. Moreover, this would be excessively expensive to be accomplished. Also exposing a near skin vein is not feasible, besides the much larger laser power needed, due to the extremely long time that would require to let all the CTCs circulating in the blood to pass in a single vein (this would not guarantee the complete elimination of all CTCs).
Minor points:
Line 66: "chemiphysical" should be corrected with chemo physical.
Line 82: it is clear that 1010 W/m2 is an error.
Line 129: "complete" should be replaced with completed.
Line 142: Figure 2, not Figure 1.
Line 147: please correct the typo (acetylglucosyamine).
Line 148: then instead of "the".
Lines 233 and 252: please correct "NH4OH" with NH4OH.
Table 1: please correct the typo (keratinocyt).